# Metagenomics Analysis of Breast Microbiome Highlights the Abundance of Rothia Genus in Tumor Tissues

**DOI:** 10.3390/jpm13030450

**Published:** 2023-02-28

**Authors:** Souad Kartti, Houda Bendani, Nasma Boumajdi, El Mehdi Bouricha, Oumaima Zarrik, Hajar EL Agouri, Mohamed Fokar, Youssef Aghlallou, Rachid EL Jaoudi, Lahcen Belyamani, Basma Elkhannoussi, Azeddine Ibrahimi

**Affiliations:** 1Biotechnology Lab (MedBiotech), Bioinova Research Center, Rabat Medical & Pharmacy School, Mohammed V University in Rabat, Rabat 10100, Morocco; 2Mohammed VI Center for Research & Innovation (CM6), Rabat 10000, Morocco; 3Pathology Department, Oncology National Institute, Rabat Medical and Pharmacy School, Mohammed Vth University in Rabat, Rabat 10100, Morocco; 4Center for Biotechnology and Genomics, Texas Tech University, Lubbock, TX 79409, USA; 5Institute of Cancer Research, Fez 30000, Morocco; 6Emergency Department, Military Hospital Mohammed V, Rabat 10000, Morocco; 7Mohammed VI University of Health Sciences, Casablanca 20000, Morocco

**Keywords:** breast cancer, metagenomics, 16s rRNA sequencing, microbiome, rhotia

## Abstract

Breast cancer is one of the main global priorities in terms of public health. It remains the most frequent cancer in women and is the leading cause of their death. The human microbiome plays various roles in maintaining health by ensuring a dynamic balance with the host or in the appearance of various pathologies including breast cancer. In this study, we performed an analysis of bacterial signature differences between tumor and adjacent tissues of breast cancer patients in Morocco. Using 16S rRNA gene sequencing, we observed that adjacent tissue contained a much higher percentage of the Gammaproteobacteria class (35.7%) while tumor tissue was characterized by a higher percentage of Bacilli and Actinobacteria classes, with about 18.8% and 17.2% average abundance, respectively. Analysis of tumor subtype revealed enrichment of genus Sphingomonodas in TNBC while Sphingomonodas was predominant in HER2. The LEfSe and the genus level heatmap analysis revealed a higher abundance of the Rothia genus in tumor tissues. The identified microbial communities can therefore serve as potential biomarkers for prognosis and diagnosis, while also helping to develop new strategies for the treatment of breast cancer patients.

## 1. Introduction

Breast cancer (BC) is the leading cancer in women in both developed and developing countries and accounts for 11.7% of all cancers [1]. It is not only the most frequently diagnosed cancer in women worldwide, but it also causes the most deaths in women (15.5%) [1]. Although diagnostic and treatment advances have significantly improved survival rates for breast cancer patients, there is still much to be learned about the underlying biology of the disease. Like other cancers, the etiology of BC remains uncertain since it is a complex multifactorial disease that probably emerges from a combination of genetic and non-hereditary (environmental) factors, which contributes to the complexity of the treatment and management of this devastating disease [2]. In addition to genetic factors, such as mutations in the BRCA1 and BRCA2 genes that represent a predisposing factor for BC, multiple environmental and lifestyle components are strongly linked to it [2]. Among the non-hereditary factors, one area of research that has received considerable attention in recent years is the study of the microbiome and its potential role in cancer development and progression. The human microbiota is of considerable interest; it is defined as a mysterious treasure of the body and plays important roles in host metabolism, digestion, and immunity [3,4]. The microbiome is the collection of microorganisms that populate various parts of the human body, e.g., skin, gut, liver, and breast tissues [5,6]. These microbiota are in balance with the human body; however, microbial disturbances (dysbiosis) could contribute to the risk of developing health problems. Recent studies have shown that the microbiome may actively participate in the development of various pathologies including cancer [7]. For example, some studies have suggested that certain types of bacteria may contribute to the development of breast cancer by producing toxins or promoting inflammation, while others have suggested that certain types of bacteria may have protective effects against breast cancer [8]. With regard to BC, one of the most important roles of the human microbiome is the regulation of steroid hormone metabolism since endogenous estrogens are a very important risk factor leading to the development of BC, especially in postmenopausal women [9]. However, the association between the human microbiome and BC remains to be explored. Some gut microbiota play an important role in breast carcinogenesis by boosting antitumor immunity and immune surveillance, and by regulating estrogen levels [10,11]. To better understand the relationship between the microbiome and breast cancer, researchers have turned to microbiome analysis techniques, which identify and quantify microorganisms in various tissues. These techniques include 16S rRNA gene sequencing, metagenomic sequencing, and shotgun metagenomic sequencing, among others. Metagenomics, which refers to the non-targeted sequencing of the entire community DNA in an environment, has allowed microbiologists to profile the taxonomic composition of the human microbiome. The microbial composition of breast tissue adjacent to the tumor was compared to the breast tissue of healthy women. The adjacent normal breast tissue contained a relatively higher abundance of bacteria which are members of Bacteroidetes phylum, Comamondaceae family, and Bacillus, Staphylococcus genera [12], which may suggest a gradual change in the microbiota from a healthy state to a cancerous. However, not all studies have identified differences between the microbiota of the tumor and the adjacent normal breast tissues [11,12]. A similar microbiota in matched tumor and normal adjacent tissues could suggest a predisposition of the entire breast tissue to carcinogenesis. Despite these findings, much is still unknown about the role of the microbiome in breast cancer. Further research is needed to fully elucidate the complex relationship between the microbiome and breast cancer, and to explore the potential of microbiome-based therapies for the prevention and treatment of breast cancer. The intent of this study was to compare the microbiome of tumor and adjacent tissues obtained from the same Moroccan patients to help us identify biomarkers that may be associated with the risk of cancer occurrence. Additionally, in this study, the BC microbiome by subtype was determined. This can be used to identify a link between this microbiome and response to treatment [13,14].

## 2. Materials and Methods

### 2.1. Sampling, Patients, and Ethics

Fresh breast tumor tissues were obtained from 47 patients who were admitted to the National Institute of Oncology, Ibn Sina University Hospital of Rabat, Morocco, from December 2018 to December 2019; the patients were pathologically diagnosed with primary BC and underwent surgery for a total or partial mastectomy. Women who have received antibiotic treatment within a period of 3 months prior to recruitment were excluded from our study. Fresh tumor and adjacent normal breast tissues pairs were obtained from the same patient. The patient’s ages ranged from 27 to 78 years (mean  =  53, SD  =  12 years); we grouped patients into 3 age groups. The first group, 16 patients, is composed of ages between 49 and 58; the second one, with 15 patients, whose ages varied between 49 and 58; and the third one with 16 patients between 59 and 78 (age at diagnosis of primary BC). The adjacent normal tissue samples were obtained from surgical specimens, sufficiently away from tumor tissues (up to 5 cm), and were verified by a pathologist to be histologically free of any tumor cells or lesions. Tumor tissue samples were obtained from the cores of tumor tissues without any contamination of normal tissues. Pathological data about each patient breast tissue specimen, including hormone receptor status grade and stage of BC, were obtained from pathological reports. All patients gave their written informed consent to the study that was approved by the local Ethical Committee of the Faculty of Medicine and Pharmacy in Rabat, Morocco (No 103/17). A total of 94 samples were preserved in Allprotect Tissue Reagent (Qiagen, Hilden, Germany) and immediately stored at −20 °C until further processing. 

### 2.2. Genomic DNA Extraction from Breast Tissue

Genomic DNA from breast tissue was extracted using the QIAamp DNA Mini Kit, (Qiagen GmbH, Hilden, Germany), according to the manufacturer’s instructions. DNA was quantified using the NanoDrop 2000c Spectrophotometer (Termo Fisher Scientifc, Waltham, MA, USA) and the Qubit dsDNA HS assay kit (Life Technologies, Carlsbad, CA, USA); samples were stored at −20 °C until 16S rRNA gene library preparation.

### 2.3. Preparation of Samples and Sequencing of 16S-rRNA Amplicons

Amplification of the hypervariable V3-V4 regions of the 16 S rRNA bacterial genes was conducted in two PCR steps: a template of 5 ng/µL of DNA from each sample was used for the first PCR, which was performed using the V3-V4 region specific primers with overhang adapters attached; Table 1 contains a list of the primer sequences that were used in this work. To verify the PCR product’s size (~460 bp), 1 µL of it was analyzed on a Bioanalyzer DNA 1000 chip (Agilent, Santa Clara, CA, USA). Subsequently, Agencourt AMPure XP beads (Beckman Coulter, Brea, CA, USA) were used to purify the 16 S V3-V4 amplicons away from free primers and primer dimer species. Purification products underwent further quality and quantity controls by Bioanalyzer DNA 1000 analysis (Agilent, Santa Clara, CA, USA). The second PCR, performed as per the Nextera XT protocol (Illumina, San Diego, CA, USA), allowed the addition of the Illumina sequencing adapters and the dual-index primers, which barcoded each sample. The V3-V4 amplified regions of each patient were purified through Agencourt AMPure XP Beads (Beckman Coulter, Brea, CA, USA), quantified using the Qubit HS assay kit (Life Technologies, Carlsbad, CA, USA), and quality-assessed using a High Sensitivity Chip on the 2100 Bioanalyzer Instruments (Agilent, Santa Clara, CA, USA). However, up to 106 libraries were pooled together for sequencing. Therefore, 8 pM of denatured libraries were combined to 25% of 8 pM PhiX control and loaded into the MiSeq v3 reagent cartridge. Sequencing reactions were performed through the Illumina MiSeq System (PE 300×2) by obtaining an average read length of about 300 bp. The raw sequencing data are available in the SRA repository under the BioProject PRJNA926328.

### 2.4. Sequencing Data Analysis

Raw sequencing reads (fastq files) were analyzed using the Quantitative Insights into Microbial Ecology version 2 (QIIME2, https://qiime2.org/, accessed on 20 June 2022), a next-generation microbiome bioinformatics platform to determine the taxonomic diversity profiles of microbiota, and the standard tools/plugins provided by QIIME2 [16]. Firstly, reads were separated with barcodes according to their sample and low-quality reads were removed. The tags were generated from the reads based on the overlapping relationship. Then, those tags were clustered into operational taxonomic units (OTUs) at the commonly used 97% similarity threshold. The OTUs were identified to the lowest taxonomic level using QIIME2 and a reference dataset from SILVA 138 99% OTUs full-length sequences database [17]. The differential abundance of each OTU was analyzed using statistical analysis in R.

### 2.5. Statistical Analyses

Linear discriminant analysis of effect size (LEfSe) [18] was performed on the microbial community relative abundance data to identify biomarkers. OTUs were analyzed through Hutlab Galaxy provided through the Huttenhower lab. In order to discriminate the features, the logarithmic score of LDA was set to 2.0. To get an idea of the expression difference of each taxon between tumor and adjacent tissues, we considered the average expression of each family or class in each group (tumor/adjacent). We conducted t-test measurements on individual differences in proportional abundances of the significantly altered microbial communities between adjacent and breast tumor tissues. For all analyses, a *p* < 0.05 is considered statistically significant. Statistics and figures were computed and produced in R (version 4.1.2).

### 2.6. Enterotype Clustering

Reproducible patterns of variation in the microbiota, e.g., the proportions of major taxa, could be separated into groups; these groups have been called enterotypes [19]. The enterotype clustering is proposed as a useful method for stratifying human microbiomes. In this study, the enterotype clusters were performed following the methods previously described by the European Molecular Biology [19] and available via http://enterotype.embl.de/enterotypes.html, accessed on 18 October 2022. Both the Jensen–Shannon divergence distance (JSD) and the partitioning around medoids (PAM) clustering algorithm were used to cluster the samples based on their respective genus abundances. The Calinski–Harabasz (CH) index was used to compute the optimal number of clusters (k = 3). We used between-class analysis (BCA) to visualize results from principal component analysis. The figure was generated using the adegraphics package (1.0–16) in Rstudio.

## 3. Results

In this study, a total of 94 fresh tumor and adjacent normal BC tissues were randomly collected from 47 women. To investigate the association between age and breast cancer microbiome, we grouped patients into three age groups. The first group, 16 patients, is composed of ages between 49 and 58; the second one, with 15 patients, whose ages varied between 49 and 58; and the third one with 16 patients between 59 and 78. We analyzed BC tissues by the four major breast tumor subtypes: luminal A, luminal B, human epidermal growth factor receptor 2 (HER2), and triple negative breast cancer (TNBC). Of the 47 BC tissues, 22 were Luminal B, 14 were Luminal A, 7 were TNBC, and 4 were HER2. Tumor receptor status was not available for six of the BC tissue specimens. The standard protocol was used for the preparation of an Illumina library to allow for the sequencing of the 16S V3-V4 amplicon. Based on age groups, we computed the individual difference in the relative abundance of the genus level between adjacent and breast tumor tissues as represented in Appendix A. The Sporosarcina genus was significantly higher in group 3 for both tumor and adjacent tissues compared to group 1 (*p*-value = 0.035). This indicated that Sporosarcina is abundant in older patients. 

The microbial signatures were investigated by the breast microbial composition of the tumor and the adjacent tissues groups. This composition differed between the two groups at several taxonomic levels (Figure 1A,B). The most predominant phyla in the two groups are, by order, Proteobacteria (class Gammaproteobacteria), Firmicutes (class Bacilli), and Actinobacteria (class Actinobacteria). Adjacent tissue contained a much higher percentage of the Gammaproteobacteria class (35.7%) while tumor tissue was characterized by a higher percentage of Bacilli and Actinobacteria classes, with about 18.8% and 17.2% average abundance, respectively. Firmicutes, Proteobacteria, and Actinobateria phyla predominated in the majority of breast bacterial community [10,20]. At the family level, Moraxellaceae dominated in tumor and adjacent tissues with the mean values 19.67% and 22.32%, respectively. Furthermore, the family Micococcaceae was detected with a percentage of 13.5% and 9.8% in tumor and adjacent tissues, respectively. The family Enterobacteriacea, of which E. coli is a member, is higher in adjacent tissue with a value of 10.7%, compared to 5.5% in tumor tissues, whereas Staphylococcaceae family was abundant in tumor tissue (9%). 

We subsequently explored the microbial composition at genus level in tumor tissues (Figure 2). According to the results, the genus Psychrobacter, Streptococcus, Acinetobacter, and Corynebacterium are present in the tumor tissue of more than 80% of breast cancer patients considered in this analysis.

To further highlight the microbial markers of the different tissue groups, linear discriminant analysis (LDA) effect size (LEfSe) was used to identify critical microbial markers of tumor tissue and adjacent tissues. Based on LEfSe analysis, the Pseudomonadace family and the Pseudomonas genus were identified to be more enriched in adjacent tissue while Peptostreptococcales _Tissierella family and Finegoldia, Rothia genus were enriched in tumor tissue (Figure 1C). 

To better capture the differences between the two groups, we compared the individual bacterial compositions at different levels (phylum, class, order, family, and genus). The adjacent tissue was significantly enriched at all levels compared to tumor tissues. The most abundant bacteria were Bdellovibrionota phylum, Oligoflexia class, Enterobacterales order, Enterobacteriaceae family, and uncultured 0319-6G20 bacteria at genus level (Figure 3).

According to the t-test results, some significant differences were observed between the tumor and adjacent tissues for the four subtypes (Figure 4A). The genus Alloiococcus was more expressed in the tumor tissues in the group luminal B (*p*-value = 0.02). Regarding the group luminal A, the genus Corynebacterium was more abundant in tumor tissues (*p*-value = 0.0024), and Lawsonella was more observed in adjacent tissues (*p*-value = 0.041). For the Basal group, a significant difference was observed in Sporosarcina genus, which is abundant in the adjacent tissues (*p*-value = 0.032). No statistically significant differences were observed at the genus level in HER2 group. Furthermore, LEfSe analysis showed that luminal A and luminal B groups had no microbial markers, whereas HER2 was abundant in genus Thermus and TNBC was abundant in family Sphingomonadaceae (Figure 4B). 

Subsequently, we generated a heatmap to visualize the genus level in tumor and adjacent tissue (Figure 5). This analysis showed that Streptococcus, Rothia, and Staphylococcus genus were more abundant in tumor tissue than in the adjacent one, which was more abundant in Escherichia-shigella genus. To group the tumor and the adjacent BC samples into enterotype clusters, the PAM method using JSD for the relative abundance of genera was used. The CH index indicated the optimal number of enterotypes as three. Principal component analysis (PCA) was used to cluster the samples of the two groups into three enterotypes. Enterotype 1 which was mainly driven by the genus Rothia, enterotype 2 by Escherichia-shigella, and enterotype 3 by the genus Psychrobacter (Figure 6).

## 4. Discussion

Previously, the breast was thought to be sterile, but today, several studies have confirmed the existence of microbes in the breast [21]. Breast tumors have a higher load and a diverse microbiome compared to other tumor types (lung, melanoma, pancreas, ovary, bone, and glioblastome) [22]. This microbiome existing in the breast environment may potentially have effects on carcinogenesis, diagnosis, and therapy of BC, which until now have not been sufficiently understood. This discovery of the breast microbiome has important implications for breast health and disease. Researchers have suggested that the breast microbiome may play a role in breast cancer development and progression, as well as other breast conditions such as mastitis [23]. For example, a study found that the presence of certain bacteria in the breast microbiome was associated with an increased risk of developing breast cancer [24]. The present study analyzed the bacterial microbiome of the tumor and their relative adjacent tissues from patients with BC in Morocco. We also compared this microbiome according to the main subtypes of BC (luminal A, Luminal B, HER2, and TNBC). The main phyla and classes of breast microbiota are Proteobacteria (class Gammaproteobacteria), Firmicutes (class Bacilli), and Actinobacteria (class Actinobacteria). Proteobacteria and Firmicutes were reported to be positively associated with adiposity [25], the occurrence of these phyla in breast tissue could be due to fatty acid richness of this environment. Proteobacteria and bacilli showed a higher relative abundance in metastatic and primary tumor [26]. The Thermus genus, thermophilic bacteria [27] belonging to the Deinococcota phylum, was abundant in HER2 subtype. In several studies, this subtype showed a high TNF-, a pro-inflammatory cytokine closely related to oxidative stress responses and cancer progression [28,29]. A relevant recent study on chickens showed that the abundance of Thermus was related to low-body-weight chickens, which had significantly upregulated inflammatory cytokines in the jejunum [30]. We speculate that the effect of Thermus in the breast cancer HER2 subtype is related to inflammatory responses. On the other hand, it turned out that HER2 subtype is related to tissue from advanced stage of lung cancer patients [26]. 

We found that the Sphingomonadaceae family was more abundant in TNBC. This family was described previously to have benefits, such as breaking down aromatic hydrocarbons as well as polycyclic aromatic hydrocarbons, both of which are known to be associated with breast cancer [31,32]. However, the ability of this taxa to metabolize aromatic hydrocarbons may confer protective properties against BC, as they were only observed in nipple aspirate fluid of healthy breast tissue. Another study has also suggested that Sphingomonadaceae may have beneficial effects on the human immune system. Furthermore, higher levels of this bacteria in the gut microbiota were associated with reduced risk of allergic sensitization in children [33]. Another study found that Sphingomonadaceae were among the bacterial families that were enriched in the gut microbiota of healthy individuals, and that the presence of these bacteria was associated with improved gut barrier function [34].

The LEfSe and the genus level heatmap analysis revealed higher abundance of Rothia genus in tumor tissues. A previous study showed that Rothia abundance in fecal samples of breast patients was negatively correlated with the level of the gut amino acid metabolite norvaline [35]. In the same study, they used L-norvaline, a type of norvaline, to investigate the inhibitory effect of L-norvaline on BC 4 T1 cells, as well as the relation between the L-norvaline concentration and the arginase1 level (Arg-1). Their results showed that by adding the doxorubicin hydrochloride (DOX) group to L-norvaline, the proliferation of BC cells could be significantly inhibited. Additionally, the level of Arg-1 decreases with high concentrations of L-norvaline, which functions as an Arg-1 inhibitor. Expression of Arg-1 leads to under-expression of p-AKT, resulting in deactivation of the AKT signaling pathway in BC cells; therefore, arg-1 acts as a tumor suppressor in BC [36]. The relationships between the abundance of Rothia genus, the level of norvaline, and the level of Arg-1 in breast tissue tumor requires further studies to understand the mechanism of the influence of Rothia and the metabolite norvaline on tumor growth. 

Using BCA from the breast microbiome of the 94 datasets, the clustering analysis showed three optimal enterotypes, characterized by higher level of Rothia, Escherichia-shigella, and Psyhcrobacter, respectively. Escherichia-shigella, a member of the Proteobacteria phylum, harbors the β-glucuronidase enzymes whose activities block the conjugation of estrogens, along with other compounds, and leaves them as biologically active hormones, consequently, the elevated levels of circulating estrogens and its metabolites, thus increasing the risk of BC [37,38,39]. This insight into the microbial community within the tumor and the adjacent BC tissues may provide fundamental information for future investigations. For example, the role of Themus and Rothia genus in breast cancer, especially in the inflammation and the breast tumor growth, remains unknown. The impact of those genus on the prognosis of patients requires further investigation. 

## Figures and Tables

**Figure 1 jpm-13-00450-f001:**
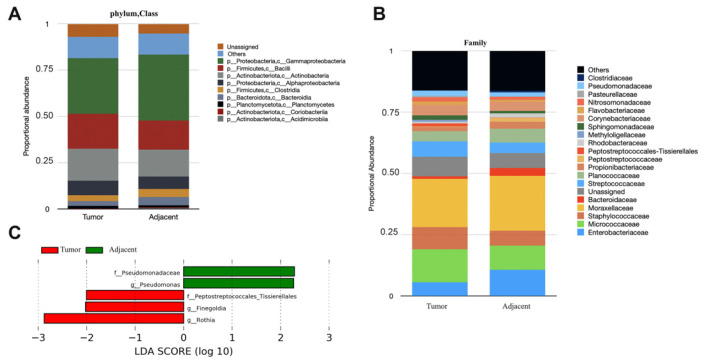
Breast microbiota is distinct between tumor and adjacent breast tissues. (**A**) Taxonomic profiles of adjacent (*n* = 47) and tumor pair (*n* = 47), microbiota at class level, and (**B**) family level for taxa with a difference in relative abundance between tumor and adjacent tissues >0.038 are shown. (**C**) Significant biomarkers between the tumor and adjacent tissues performed by linear discriminant analysis effect size (LEfSe) scores. A negative LDA score represents the tumor group (shown as red). Abbreviations: p—phylum; c—class; o—order; f—family, and g—genus.

**Figure 2 jpm-13-00450-f002:**
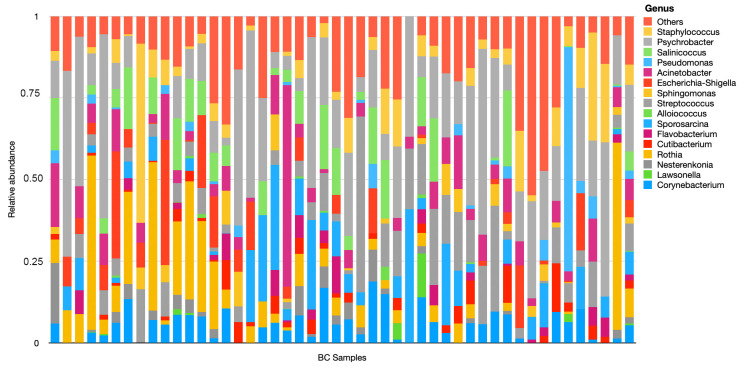
Relative microbial abundance at genus level in tumor tissues. BC Samples: breast cancer samples.

**Figure 3 jpm-13-00450-f003:**
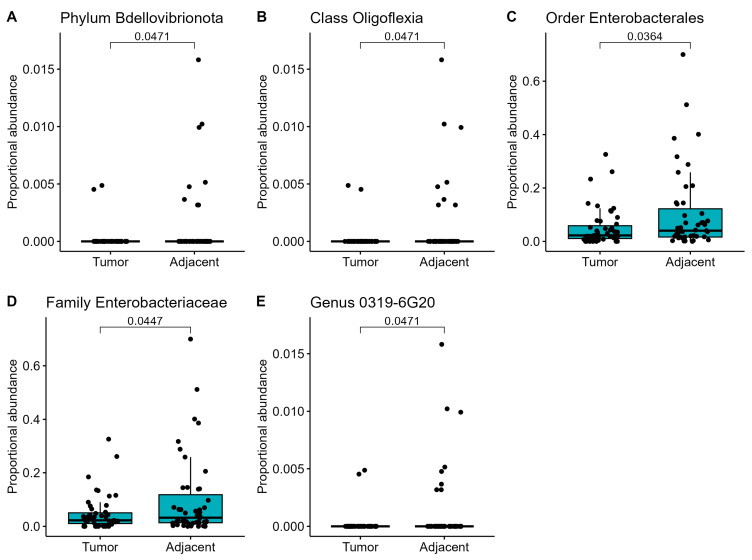
(**A**–**E**) Individual differences in proportional abundances of the most significantly altered microbial communities between adjacent and breast tumor tissues at different levels. A *p* < 0.05 is considered statistically significant based on t-test measurements.

**Figure 4 jpm-13-00450-f004:**
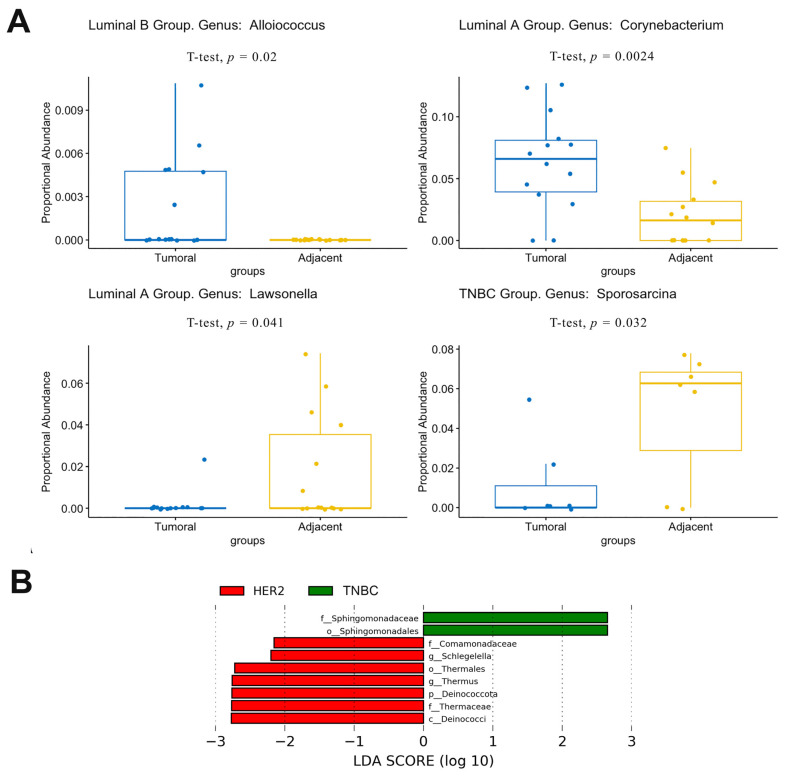
(**A**) Individual differences in proportional abundances of significantly altered microbial genus between adjacent and breast tumor tissues in the hormonal patients’ groups. A *p* < 0.05 is considered statistically significant based on t-test measurements. (**B**) LDA score computed on the microbial relative abundance between breast tumor subtypes. The TNBC group is shown as green, and the HER2 group as red.

**Figure 5 jpm-13-00450-f005:**
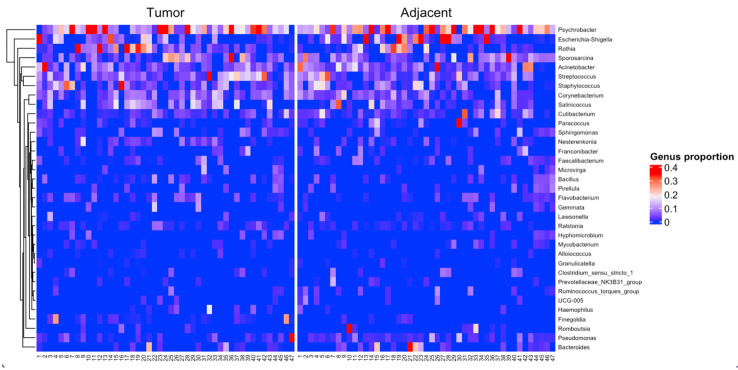
Heatmap illustrating levels of genus in breast by sample grouped into tumor and adjacent samples. The rare genus is shown in the blue color while the red color represents abundant genus. The dendogram shows the hierarchical clustering of genus.

**Figure 6 jpm-13-00450-f006:**
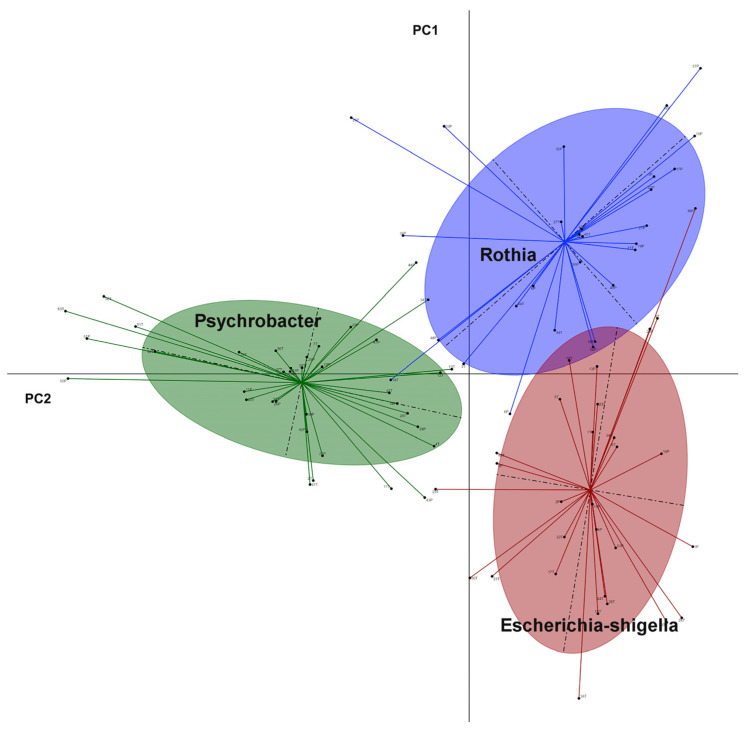
Enterotypes identified in the 94 breast cancer samples (tumor and adjacent tissues) using principal coordinate analysis (PCA). The blue cluster is enterotype 1 (Rothia), the green one represents enterotype 2 (Psychrobacter), and enterotype 3 is shown in brown (Escherichia-shigella).

**Table 1 jpm-13-00450-t001:** Primers used to amplify the V3-V4 regions encoding for the 16 S rRNA for sequencing library preparations.

16s Region	Name	F/R	Sequence
**V3-V4 [15]**	S-D-Bact-0341-b-S-17	F	5′ TCG TCG GCA GCG TCA GAT GTG TAT AAG AGA CAG CCT ACG GGN GGC WGC AG
	S-D-Bact-0785-a-A-21	R	5′ GTC TCG TGG GCT CGG AGA TGT GTA TAA GAG ACA G GAC TAC HVG GGT ATC TAA TCC

## Data Availability

The raw sequencing data are available in the SRA repository under the BioProject PRJNA926328.

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
