# Peer review of "Metagenomics Analysis of Breast Microbiome Highlights the Abundance of Rothia Genus in Tumor Tissues"

_jpm, 2023, doi:10.3390/jpm13030450_

Round 1

Reviewer 1 Report

The manuscript is well-written and scientifically sound but I have a few queries:

1. Why did the author avoid discussing the risk factor while discussing disease prognosis? Age factor is crucial for any complex disease including breast cancer. Therefore it is required to represent the patient's age distribution instead of the range because there is a massive difference in the age of study subjects. 

2. How could you justify the age difference between patients?

3. What are inclusion and exclusion criteria and guidelines for study subjects?

4. Figure 3A must have a proportional abundance of altered microbial in HER2 group too. The author should keep it that data should not be overestimated and in LDA plats or as per the author's choice.

5. This is a suggestion to represent data in the genus of abundance microbial in BC group wised. Which is a more simple way to attract readers.  

Author Response

Response to Reviewer 1 Comments

We appreciate the time and effort you have put into reviewing our manuscript entitled: “Metagenomics analysis of breast microbiome highlights the abundance of Rothia genus in tumor tissues” and writing relevant comments that significantly improve our paper. These changes are highlighted in the manuscript in red. Below, in red, are the point-by

point responses to the reviewers' comments.

Point 1: Why did the author avoid discussing the risk factor while discussing disease prognosis? Age factor is crucial for any complex disease including breast cancer. Therefore it is required to represent the patient's age distribution instead of the range because there is a massive difference in the age of study subjects. 

Response 1: Thank you for this suggestion which will enrich our discussion. Indeed, we represented the age of patients as a distribution(lines 66-68). Furthermore, we conducted t-test measurements on proportional abundances of the most significantly altered microbial genus in adjacent and breast tumor tissues between the three patient's age distribution groups (Figure 1S). we reported results at lines 149-151.

Point 2:  How could you justify the age difference between patients?

Response 2: There were no criteria concerning the age of patients while choosing the study subjects. Over a period of 12 months, we considered only inclusion and exclusion criteria.
The age of the patients varied between 27 and 78 years.

Point 3:  What are inclusion and exclusion criteria and guidelines for study subjects?

Response 3: We recruited women diagnosed with incident breast cancer who underwent surgery and we excluded women who have received antibiotic treatment within a period of 3 months prior to recruitment (We have added more information in the materials and methods section, lines 63-66). Thank you for drawing our attention to this important point.

Point 4:  Figure 3A must have a proportional abundance of altered microbial in HER2 group too. The author should keep it that data should not be overestimated and in LDA plats or as per the author's choice.

Response 4: Data described in Figure 3A concern only significant results, as mentioned in figure’s legend, no statistically significant differences were observed at the genus level in HER2 group (Line 205-206 at results section).

Point 5:  This is a suggestion to represent data in the genus of abundance microbial in BC group wised. Which is a more simple way to attract readers.  

Response 5: Thank you for your suggestion, based on your comment, we added a representation of genus abundance in BC group (Figure 2).

Reviewer 2 Report

The authors performed a comprehensive analysis for breast microbiome from breast tumors and adjacent tissues. Overall, the study is pretty interesting. Some issues below should be addressed.

1). There are no any discussion or author's thoughts in the result part (although there have a discussion in the end, but very few and not informative). For those identified differentiated microbiome, the authors should read more literatures about their well-known function and propose some hypothesis based on their own data, not just describing their data.

2) The authors only compare the tumors with corresponding adjacent tissues. It would be interesting to compare the differences of microbiome between different types of tumors. These may provide some new insights between microbiome and cancer types.

Author Response

We appreciate the time and effort you have put into reviewing our manuscript entitled: “Metagenomics analysis of breast microbiome highlights the abundance of Rothia genus in tumor tissues” and writing relevant comments that significantly improve our paper. These changes are highlighted in the manuscript in red. Below, in red, are the point-by

point responses to the reviewers' comments.

Point 1: There are no any discussion or author's thoughts in the result part (although there have a discussion in the end, but very few and not informative). For those identified differentiated microbiome, the authors should read more literatures about their well-known function and propose some hypothesis based on their own data, not just describing their data.

Response 1: Thank you, based on your suggestion bibliographic references have been consulted, and the discussion was amended (lines 260-268)

Point 2:  The authors only compare the tumors with corresponding adjacent tissues. It would be interesting to compare the differences of microbiome between different types of tumors. These may provide some new insights between microbiome and cancer types.

Response 2: We definitely agree with the reviewer and it would be interesting to compare the differences of microbiome between different types of tumors and cancers. Indeed and since the aim of this work was to study the impact of the Microbiome on breast cancer, it is absolutely logical to extend the study to other tumors and cancers as insightful perspectives.

Reviewer 3 Report

Kartti et al. analyzed the microbiome composition of tumor and adjacent normal tissues from breast cancer patients in Morocco. They discovered the difference in microbiome composition between tumor and adjacent tissues as well as between breast cancer tumor subtypes. For example, they found that the Sphingomonadaceae family is more abundant in triple negative breast cancer TNBC. This family was found to be beneficial against breast cancer by previous studies. The enrichments of other microbiome genus or family were also identified in tumor tissues or different breast cancer subtypes. This study will help study and understand how the difference in microbiome composition is related to breast cancer metabolism and development. This paper is good for publication after minor revision.

Minor comments:

1.      Figure 3B, the legend is confusing. The coloring of the tumor subtypes should be “red as HER2 and green as TNBC”.

2.      Figure 4, It is good to have the representation of the samples in numerical order. A high-resolution image will be better.

Author Response

Response to Reviewer 3 Comments

We appreciate the time and effort you have put into reviewing our manuscript entitled: “Metagenomics analysis of breast microbiome highlights the abundance of Rothia genus in tumor tissuesand writing relevant comments that significantly improve our paper.  These changes are highlighted in the manuscript in red. Below, in red, are the point-by

point responses to the reviewers' comments.

Point 1:  Figure 3B, the legend is confusing. The coloring of the tumor subtypes should be “red as HER2 and green as TNBC”.

Response 1: Thank you for bringing this to our attention, we rectified the legend of Figure 4B. (lines 213-214)

Point 2:  Figure 4, It is good to have the representation of the samples in numerical order. A high-resolution image will be better.

Response 2: As you suggested, we ordered the samples in numerical order and improved the resolution of Figure 5 (previously Figure 4)